# Prevalence of HTLV-1 and Hepatitis B Surface Antigen (HBsAg) Positivity among MSM Attending a Large HIV Treatment Centre in Trinidad

**DOI:** 10.3390/v16071169

**Published:** 2024-07-20

**Authors:** Robert Jeffrey Edwards, Selena Todd, Jonathan Edwards, Noreen Jack, Gregory Boyce

**Affiliations:** Medical Research Foundation of Trinidad and Tobago, 7 Queen’s Part E, Port-of-Spain 150123, Trinidad and Tobago

**Keywords:** HBsAg, HTLV-1, HIV-1, coinfection, MSM

## Abstract

HIV-1, Hepatitis B and HTLV-1 have similar risk factors and shared routes of transmission and MSM are disproportionately affected by HIV. The aim of the study was to determine the prevalence of HTLV-1 and HBsAg positivity at initial enrolment among MSM attending a large HIV Clinic in Trinidad. Chart reviews were conducted between 2 and 15 January 2024, among self-identified MSM and a comparative group of randomly selected self-identified heterosexual males where sociodemographic, clinical and laboratory data were collected and analysed using SPSS Version 25. During the period April 2002–31 October 2023, in total there were 10,424 patients registered at the clinic, of whom 1255 (12.0%) were self-identified MSM, with an age range of 19–85 years and a median age of 40 years. There were 1822 randomly selected heterosexual males, with an age range of 18–94 years old and a median age of 52 years. Among the MSM, there were 21 (1.67%) patients who were HIV-1/HTLV-1-coinfected, 64 (5.10%) who were HIV-1/HBsAg-coinfected and two (0.16%) who were coinfected with all three viruses (HIV-1/HTLV-1/HBsAg) as compared to 47 ((2.58%) HIV-1/HTLV-1-coinfected (*p* = 0.12), 69 (3.79%) HIV-1/HBsAg-coinfected (*p* = 0.10) and three (0.16%) patients coinfected with all three viruses among the heterosexual males. There were no patients with HTLV-1-related diseases among the HIV-1/HTLV-1-coinfected patients and there were no deaths from chronic liver disease in patients coinfected with HIV-1/HBsAg. Despite the availability of an efficacious vaccine, there is a prevalence of hepatitis B of 5.1% among MSM attending the HIV Clinic in Trinidad; therefore, programmes to increase health literacy, screening and immunization are urgently needed.

## 1. Introduction

In 1980, Poiesz et al. isolated HTLV-1 in a cell-line from a patient with cutaneous T-Cell lymphoma and HTLV-1 was the first retrovirus to be identified [1]. HTLV-1 is endemic in sub-Saharan Africa, South America, the Caribbean, southwest Japan, central parts of Australia, Melanesia and the Middle East [2,3] but it is still considered a neglected disease that requires prioritization by public health officials [4]. Globally, it is estimated that approximately 5–10 million people may be infected with HTLV-1 [3] and a systematic review and metanalysis conducted by Schierhout et al. [5] showed that, compared with HTLV-1 negative persons, the adjusted risk of death from any cause was higher in persons infected with HTLV-1 (RR 1·57, 95% CI 1.37–1.80). Approximately 5–10% of infected patients develop a number of HTLV-1-associated diseases including adult T-cell leukemia/lymphoma (ATLL) [1,6], tropical spastic paraparesis/HTLV-1-associated myelopathy (TSP/HAM) [3,7], infective dermatitis [8] and uveitis [9].

HTLV-1 is transmitted from mother to child, mostly through prolonged breastfeeding [10,11], via sexual intercourse [12] and parenterally via sharing of needles/syringes and transfusion of infected blood and blood products [2]. The seroprevalence of HTLV-1 is reported to be age- and sex-dependent, with an increase in prevalence with age and a higher prevalence in women [13]. Edwards et al. [14] conducted a retrospective study to determine the seroprevalence of HTLV-1 coinfection at the initial visit among patients attending the HIV clinic in Trinidad over the period of April 2002–December 2018. There were 8916 patients enrolled in the clinic over the study period, 159 were HIV-1/HTLV-1 coinfected, with 87 (54.7%) females and 72 (45.3%) males with an HTLV-1 seroprevalence of 1.78% [14]. There is a lack of data on the prevalence of HTLV-1 infection among MSM in the Caribbean but in Central Brazil, Castro et al. [15] reported an HTLV-1 prevalence of 0.7% in MSM, which is similar to that seen among blood donors in Brazil [15] and La Rosa et al. [16] in Peru in 2002–2003 reported a prevalence of HIV-1/HTLV-1 coinfection of 4.6% among 329 HIV-positive samples from MSM.

Hepatitis B is a DNA virus that belongs to the *hepadnaviridae* family [17]. In endemic areas, it is mainly transmitted from mother to child at birth and via horizontal transmission (exposure to infected blood) among household contacts. Hepatitis B is also transmitted via exposure to blood and blood products and via sexual transmission [17]. A systematic analysis for the Global Burden of Disease (hepatitis B) Study estimated that in 2019, globally, there were 316 million persons living with chronic hepatitis B with an all-age HBsAg prevalence of 4.1% and an all-age HBsAg prevalence of 1.2% in the region of the Americas [18]. Globally, in 2019, there were an estimated 550,000 deaths from HBV-related diseases [18]. There are few reported studies of HIV-1/HBsAg coinfection in MSM living with HIV, a study conducted in 12 cities in India over the period 2012–2013 demonstrated an HBsAg prevalence of 8% in MSM living with HV [19], and a study conducted in Abuja and Lagos Nigeria among MSM and transgender women living with or at high risk for HIV showed the prevalence of HIV-1/HBV coinfection was 6% [20]. Hepatitis B is a vaccine-preventable disease and was the first anti-cancer vaccine (prevents hepatocellular carcinoma) developed [17]. The genetically engineered DNA recombinant vaccine received approval from the US Food and Drug Administration (FDA) in 1986 and, since then, has been in widespread use [21].

The twin-island republic of Trinidad and Tobago (T&T) is the most southerly of the islands of the Caribbean chain and has a population of approximately 1,367,510 persons (2023 mid-year estimate). The multi-racial population is comprised of persons of East Indian descent (35.4%), persons of African ancestry (34.2%), those of mixed race (23.8%), and 8.4% of other ethnic groups (European, Asian, Middle Eastern). Blattner et al. [22] screened a 1982 population-based hepatitis serosurvey of 1581 persons in Trinidad for hepatitis B antibodies and HTLV-1 antibodies and determined the seroprevalence of HTLV-1 was 2.2% and hepatitis B antibodies were 19.1% [22]. Among persons of African origin, the HTLV-1 seroprevalence was 3.2% (33/1025) and 0.2% (1/487) among persons of East Indian origin [20], displaying a racial disparity [22].

The first cases of AIDS in Trinidad and Tobago were reported in 1983 [23] among MSM and, in 1985, there was a transition to mainly heterosexual transmission of HIV [24]. In 2002, antiretroviral therapy (ART) became available at no cost to persons living with HIV (PLHIV) [25] funded by the government [26] and it is estimated that in 2022 there were 12,000 PLHIV in Trinidad and Tobago with an adult HIV prevalence of 1% [27] and approximately 60% of these persons are receiving ART [27].

HIV-1, hepatitis B and HTLV-1 have similar risk factors and shared routes of transmission and MSM are disproportionately affected by HIV due to increased susceptibility of the rectal mucosa to HIV acquisition and transmission [28,29], increased number of lifetime partners [28.29], role versatility [29], high-risk sexual networks [29], disinhibited substance use [30], condomless anal sex [28,29], frequent asymptomatic sexually transmitted infections (STIs) with resulting inflammation and ulceration [31,32], internalized stigma that may result in depression and other mental health disorders [29,30] and perceived and experienced discrimination by healthcare providers [33]. The purpose of the study was to determine the prevalence of HTLV-1 and HBsAg positivity among self-identified MSM and a comparative group of heterosexual males living with HIV at the initial visit to the HIV Clinic in Trinidad with a view to understanding the burden of these coinfections so that additional prevention and control strategies may be implemented.

## 2. Methodology

The Medical Research Foundation of Trinidad and Tobago (MRFTT) is the largest HIV Treatment Centre in the English-speaking Caribbean and as of 31 October 2023, there were a total of 10,424 patients registered at the clinic. Of these, 1255 (12.0%) were self-identified MSM. As part of the routine standard of care at the HIV Clinic, at the initial visit, data are collected on all patients using the routine pre-designed pro-forma surveillance form and included demographic data, ethnicity, sexual orientation, education and employment status and patients were offered screening tests for HTLV-1 antibodies, hepatitis B surface antigen (HBsAg). The CD4+ cell counts and HIV viral loads were obtained. From 2002 to 2012, the Vironostika HTLV-1 enzyme-linked immuno-assay (ELISA) (Vironostika, Organon Teknika, Durham DC) was used as the screening ELISA test, which was then discontinued and followed by the use of the HTLV-1/2 Ab Diapro ELISA (Dia.Pro, Milan, Italy). No Western blot confirmation was performed on the HTLV-1-positive samples over the period 2002–2018 and sera were frozen and stored, awaiting Western blot confirmation. All clients with ELISA results seropositive for HTLV-1 over the period 2002–2012 were re-screened with the HTLV-1/2 Ab Diapro ELISA (Dia.Pro, Milan, Italy) so that there was consistency in the ELISA screening. All positive ELISA samples over the period April 2002-October 2023 were confirmed in batches from 2018 onwards using the Western blot (MP Diagnostic ^TM^ HTLV BLOT 2.4 Western Blot Assay, MP Biomedicals Asia Pacific Pte. Ltd., Singapore, Singapore), which is able to differentiate antibodies to HTLV-1 and HTLV-2. There were three samples that were positive using ELISA and indeterminate by Western blot (WB). These patients were re-screened by ELISA and WB approximately 6–12 months after initial sample collection and still remained WB indeterminate, so these patients were excluded from the study.

Testing for HBsAg was conducted using the enzyme-linked immuno-assay (ELISA) tests ElAgen HBsAg (Adaltis Srl, Rome, Italy) over the period 2002–2018 and from 2018 to 2023 the enzyme-linked immuno-assay (ELISA) tests: Dia.Pro HBsAg^®^ one version ULTRA (Diagnostic BioProbes Srl, Milan, Italy) was used.

### 2.1. Inclusion Criteria

Age 18 years and older;Patients living with HIV;Patient attending the MRFTT for HIV treatment and care;Self-identified as heterosexual, homosexual, bisexual or other MSM.

### 2.2. Exclusion Criteria

Age under 18 years old;HIV seronegative individuals.

Testing for HIV-1 was performed using Alere Determine^®^ HIV-1/2 Ab rapid test (Alere Inc., Waltham, MA, USA) and the Uni-Gold^®^ Recombigen HIV-1/2 Ab rapid test (Trinity Biotech PLC, Bray, Ireland) and all positive samples were confirmed by enzyme-linked immunosorbent assay (ELISA) testing in accordance with the National HIV Testing Algorithm.

To determine if there are differences in the prevalences of HTLV-1 and Hepatitis B between self-identified heterosexual males and MSM, a power of 90%, a significance level of 0.05 and an estimated prevalence of 3% for hepatitis B for heterosexual males [34] were used. It was calculated that 1822 heterosexual males would be required for the comparator group to ensure that the study was adequately powered to detect a true difference if it exists.

An electronic medical records system named CELLMA was used at the MRFTT. A randomly selected comparison group of 1822 HIV-1 infected, self-identified heterosexual males enrolled at the clinic over the same period (April 2002–31 October 2023) was drawn from the entire population of HIV-1 infected, self-identified, heterosexual male patients at the clinic via computer-generated random numbers using their unique clinic identification numbers. The randomised list of HIV-1 infected, self-identified, heterosexual male patients was then input into CELLMA, and a chart review study on these patients was carried out.

A list of HIV-1/HTLV-1 and HIV-1/HBsAg coinfected self-identified MSM who ever attended the clinic during the period April 2002–31 October 2023 and a list of HIV-1/HTLV-1 and HIV-1/HBsAg coinfected self-identified heterosexual males (from the randomized list of heterosexual males) was generated using CELLMA and retrospective chart review studies were conducted on both groups of patients over the period 2–15 January 2024.

### 2.3. Ethical Approval

The study and related study procedures were reviewed and approved by the Campus Research Ethics Committee, University of the West Indies, St Augustine, Trinidad, approval number CREC-SA.2458/12/2023.

### 2.4. Data Analysis

Data were abstracted from client records to obtain the sociodemographic, clinical records and the laboratory diagnosis of HTLV-1 and hepatitis B. SPSS Version 25 was utilised for statistical analysis. The prevalence of HIV-1/HTLV-1 and HIV-1/HBsAg coinfection was determined by dividing the number of patients who had been given a diagnosis of HIV-1/HTLV-1 and HIV-1/HBsAg coinfection, respectively, by the number of MSM in the clinic population and randomized heterosexual males during the study time period. Any missing data were dropped from the statistical analysis. The dataset comprised categorical variables—type (HIV-1/HBsAg vs. HIV-1/HTLV-1), ethnicity, sex, sexual orientation (heterosexual vs. homosexual/bisexual), AIDS-defining infections and numerical variables—age, CD4+ count and HIV-1 viral load. Data were analyzed using various statistical methods to explore relationships and differences within the dataset. Measures of central tendency (mean and median) summarized numerical variables like age, CD4+ count, and HIV-1 viral load. Chi-square tests assessed associations between categorical variables such as infection type (HIV-1/HBsAg vs. HIV-1/HTLV-1), ethnicity, sex, sexual orientation (heterosexual vs. homosexual/bisexual), and AIDS-defining infections. Fisher’s Exact Test was used when expected frequencies were low. The Mann–Whitney U Test (Wilcoxon Rank Sum Test) compared numerical variables between two groups, providing a non-parametric alternative for data that did not follow a normal distribution. This test was used to compare variables such as age, CD4+ count, and HIV-1 viral load across different groups based on sexual orientation and infection type. A *p*-value < 0.05 was considered statistically significant.

## 3. Results

During the period April 2002–31 October 2023, there were 10,424 patients registered at the HIV MRFTT Clinic, with an age range of 18–97 years, and a median age of 48.1 years, 5667 (54.4%) males and 4757 (45.6%) females) and there were 1255 (12.0%) patients who self-identified as MSM.

Table 1 shows the baseline characteristics of the 1255 MSM study patients at the initial visit over the period April 2002–October 2023, with an age range of 19–85 years, and a median age of 40 years. Of these, 681 (54.3%) were of African origin, 144 ((11.5%) were of East Indian origin, 283 (22.5%) were of mixed race and 31 (2.5%) were persons of other ethnicities (Chinese, Caucasian, Lebanese). There were no available data on ethnicity for 116 (9.2%) persons.

Table 1 shows the baseline characteristics of the randomly generated 1822 self-identified heterosexual males at the initial visit over the period April 2002–October 2023, with an age range of 18–94 years, and a median age of 52 years. Of these, 1198 (65.8%) were of African origin, 139 (7.6%) were of East Indian origin, 282 (15.5%) were of mixed race and 10 (0.5%) were persons of other ethnicities (Chinese, Caucasian, Lebanese). There were no available data on ethnicity for 193 (10.6%) persons.

Among the 1255 clients who self-identified as MSM, at the initial visit, the median CD4+ and the median HIV-1 viral load were 340 cells/mm^3^ and 25,968 copies/mL, respectively, as compared to a lower median CD4+ and a higher median HIV-1 viral load of 226 cells/mm^3^ (*p* = 0.40) and 45,427 copies/mL (*p* < 0.01), respectively, in the 1822 heterosexual males (Table 1), suggesting that the heterosexual males in the study seem to have more advanced HIV disease at the initial visit.

Table 2 shows the prevalence of HTLV-1 and HBsAg coinfection by ethnicity among MSM and the randomly selected heterosexual males attending the HIV Treatment Centre in Trinidad. Among the 1255 clients who self-identified as MSM, 21 (1.67%) were HIV-1/HTLV-1 coinfected at the initial visit as compared to 47 (2.58%) of the 1822 heterosexual males (*p* = 0.12). There were no statistically significant associations between HIV-1/HTLV-1 coinfection and ethnicity among both study populations. Among the 1255 clients who self-identified as MSM, 64 (5.10%) were HIV-1/HBsAg coinfected at the initial visit as compared to 69 (3.79%%) of the 1822 heterosexual males (*p* = 0.10). MSM of African origin were more likely to be HIV-1/HBsAg coinfected as compared to heterosexual males (*p* = 0.04) of African descent.

Table 3 shows a comparison of HIV-1/HTLV-1 coinfection among heterosexual males vs. MSM patients at the initial visit. Among the 1255 clients who self-identified as MSM, 21 (1.67%) were HIV-1/HTLV-1 coinfected, with an age range of 21–68 years, and a median age of 34 years, the median CD4+ and the median HIV-1 viral load were 210 cells/mm^3^ and 46,238 copies/mL, respectively, and there were four (19.0%) AIDS-defining illnesses. In comparison, among the heterosexual males, 47 (2.58%) were HIV-1/HTLV-1 coinfected (*p* = 0.12), with an age range of 35–93 years, median age of 60 years and at the initial visit, the median CD4+ and the median HIV-1 viral load were 257 cells/mm^3^ (*p* = 0.34) and 51,250 copies/mL (*p* = 0.25), respectively, and there were 17 (36.2%) AIDS-defining illnesses (*p* = 0.06).

Table 4 shows a comparison of HIV-1/HBsAg coinfection among heterosexual males vs. MSM patients at the initial visit. Among the 1255 clients who self-identified as MSM, 64 (5.10%) were HIV-1/HBsAg coinfected, with an age range of 22–73 years, and a median age of 36 years. The median CD4+ and the median HIV-1 viral load were 353 cells/mm^3^ and 34,376 copies/mL, respectively. In comparison, among the heterosexual males, 69 (3.79%) were HIV-1/HBsAg coinfected (*p* = 0.10), with an age range of 22–77 years, and a median age of 49 years, with a lower median CD4+ and a higher median HIV-1 viral load of 208 cells/mm^3^ (*p* < 0.01) and 76,198 copies/mL (*p* < 0.01), respectively.

## 4. Outcomes

Comparing the overall outcomes of both study populations as of 31 October 2023, of the 1255 self-identified MSM, there were 764 (60.9%) in active care and follow-up; 89 (7.1%) had died; 275 (21.9%) were lost to follow-up; 84 (6.7%) migrated and 43 (3.4%) were transferred to another clinic. Of the 1822 heterosexual males in the study, there were 792 (43.5%) in active care and follow-up (*p* = 0.32); 496 (27.2%) had died (*p* = 0.02); 407 (22.3%) were lost to follow up (*p* = 0.87); 48 (2.7%) migrated (*p* = 0.21) and 79 (4.3%) were transferred to another clinic (*p* = 0.68).

There were no deaths among the HIV/HTLV-1 coinfected MSM patients, all 21 were alive as of October 2023, in treatment and care and on ART with a median CD4+ count of 449 cells/mm^3^ and a median HIV-1 viral load of <40 copies/mL. As of October 2023, of the 47 HIV-1/HTLV-1 coinfected heterosexual males, 22 (46.8%) were in active care and follow-up and on ART with a median CD4+ count of 469 cells/mm^3^ (*p* = 0.99) and a median HIV-1 viral load of <40 copies/mL (*p* = 0.26), there were 18 (38.3%) deaths (*p* < 0.01) and seven (14.9%) patients who were lost to follow-up (*p* = 0.03). There were no HIV-1/HTLV-2 coinfected clients diagnosed in the study. The causes of death in the 18 HIV-1/HTLV-1 coinfected heterosexual males included five patients with AIDS-related complications, three patients with renal failure, two patients with disseminated histoplasmosis, one patient with cerebral toxoplasmosis and seven patients with unknown causes. No patients died due to complications of chronic hepatitis.

As of October 2023, among the 64 MSM patients coinfected with HIV-1/Hepatitis B, 33 (51.6%) patients were in active care and follow-up and on ART with a median CD4+ count of 451 cells/mm^3^ and a median HIV-1 viral load of <40 copies/mL, there were 10 (15.6%) deaths and 21 (32.8%) patients were lost to follow-up. The causes of death in the 10 HIV-1/Hepatitis B coinfected MSM patients included three patients with tuberculosis, two patients with disseminated histoplasmosis, two patients with unknown causes of death, one from chronic renal failure, one from complications of AIDS, and one patient due to homicide. As of October 2023, among the 69 heterosexual patients coinfected with HIV-1/Hepatitis B, 39 (56.5%) patients were in active care and follow-up and on ART with a median CD4+ count of 434 cells/mm^3^ (*p* = 0.73) and a median HIV-1 viral load of <40 copies/mL (*p* = 0.10), there were 15 (21.7%) deaths (*p* = 0.34) and 15 (21.7%) were lost to follow-up (*p* = 0.54). The causes of death in the 15 HIV-1/Hepatitis B coinfected heterosexual males included six patients with AIDS-related complications, two patients with TB (one of whom had elevated liver function tests), one patient with chronic renal failure, one patient with disseminated histoplasmosis, one patient with pneumocystis pneumonia (PCP), one patient due to neurosyphilis, one patient due to myocardial infarction and two patients with unknown causes. No patients died due to complications of chronic hepatitis, though one patient had elevated serum transaminases but died from complications of TB.

Two of the MSM study patients were coinfected with all three viruses, HIV-1, HTLV-1 and hepatitis B; Patient #1 is a 38-year-old male of African origin whose initial CD4+ count was 561 cells/mm^3^ and HIV-1 viral load was 7949 copies/mL. He was treated for late latent syphilis in August 2021 but had no history of opportunistic infections or chronic liver disease and defaulted from care in November 2021; Patient #2 is a 52-year-old male of African origin whose initial CD4+ count was 326 cells/mm^3^ and HIV-1 viral load was 12,245 copies/mL. He had no history of opportunistic infections or chronic liver disease and is currently doing well and virally suppressed (HIV viral load < 40 copies/mL) on ART.

Three of the Heterosexual male study patients were coinfected with all three viruses, HIV-1, HTLV-1 and hepatitis B; Patient #1 is a 74-year-old male of African origin who was diagnosed with HIV in 2016, with an initial CD4+ count of 162 cells/mm^3^ and an initial HIV viral load of 920,954 copies/mL. He started ART and was doing well until his wife died in 2021 and he became depressed and defaulted from care. He had no history of opportunistic infections or chronic liver disease and was last seen in February 2024 with an HIV viral load of 40,046 copies/mL. Patient #2 was a 40-year-old male of mixed race who presented to the clinic very ill with an initial CD4+ count of 111 cells/mm^3^ and a HIV initial viral load of 1,164,235 copies/mL. He was referred to hospital and died four days later due to complications of AIDS. Patient #3 was a 40-year-old male of African origin whose initial CD4+ count was 335 cells/mm^3^, initial HIV viral load was 10,995 copies/mL and who had a serum creatinine of 11.21 mg/dL He had no history of opportunistic infections or chronic liver disease and was receiving dialysis but defaulted from care and died from the complications of renal failure.

## 5. Discussion

The HTLV-1 prevalence among 1255 MSM living with HIV attending the Clinic at the MRFTT was 1.67%, and among the randomly generated list of 1822 self-identified heterosexual males was 2.58% (*p* = 0.12), which was similar to an HTLV-1 seroprevalence of 1.78% published in 2022 among all patients attending the HIV Clinic in Trinidad [14]. The non-statistically significant lower HTLV-1 seroprevalence among MSM in the study was unpredicted as, due to high-risk sexual behaviours and increased number of lifetime sexual partners, MSM are reported to have an increased risk of acquiring HTLV-1 infection [15]. This is in contrast to studies conducted by La Rosa et al. [16] in Peru in 2002–2003, which showed a prevalence of HIV-1/HTLV-1 coinfection of 4.6% among 329 HIV-positive samples from MSM and from a study of 100 MSM attending a STI Clinic in Trinidad in 1983/84 conducted by Bartholomew et al. [35], which reported a prevalence of HIV-1/HTLV-1 coinfection of 6%. In the study by Bartholomew et al. [35], 34% of the study population were HIV-1 mono-infected and 15% were singly HTLV-1-infected; however, the sample size of MSM was small and 60% of the study population was HIV uninfected [35].

The prevalence of HIV-1/HBsAg coinfection among MSM living with HIV in the study was 5.1% and among heterosexual males was 3.79% (*p* = 0.10). The results among MSM in our study were similar to a study conducted in 12 cities in India over the period 2012–2013 among MSM living with HIV where the HBsAg prevalence was 8% [19] and a study conducted in Abuja and Lagos Nigeria among MSM and transgender women living with or at high risk for HIV where the prevalence of HIV-1/HBV coinfection was 6% [20], though the prevalence of hepatitis B is higher in India and Africa as compared to the Americas [18]. Similar results were also reported in a systematic review and metanalysis of persons living with HIV (PLHIV) in Latin America and the Caribbean where the estimated pooled prevalence of HBsAg was 7.0% in selected studies [36], which was also unexpected, as due to high-risk sexual behaviours, MSM are thought to have an increased risk of acquiring HBV infection.

The median CD4+ count at the initial visit of 210 cells/mm^3^ among the HIV-1/HTLV-1 coinfected MSM patients was slightly lower than the median CD4+ count of 257 cells/mm^3^ among the HIV-1/HTLV-1 coinfected heterosexual patients in the study (*p* = 0.34). Studies by Schechter et al. [37] and Gudo et al. [38] showed that CD4+ counts were significantly higher in the HIV-1/HTLV-1 coinfected patients [37] and other studies showed that coinfected patients sustain stable CD4+ T-cell counts that may mask the immunosuppression as patients progress to AIDS-defining illnesses and adversely affect the clinical decision making regarding opportunistic infections prophylaxis [14,38,39]. However, the median HIV viral load of 51,250 copies/mL among the HIV-1/HTLV-1 coinfected heterosexual patients was not statistically significantly higher than the median viral load of 46,238 copies/mL (*p* = 0.25) among the HIV-1/HTLV-1 coinfected MSM patients. In a study published in 2022 among PLHIV attending an HIV Clinic in Trinidad [14], the HIV-1/HTLV-1-coinfected patients had higher HIV-1 viral loads and more opportunistic infections signifying a worse prognosis, though there were no statistically significant differences in CD4+ counts as compared to the HIV-1 singly infected patients [14]. “Treat All” (where ART was started irrespective of CD4+ count) was instituted in 2017 [25,26] but before this, patients attending the HIV Clinic were offered ART if they had CD4+ counts of less than 350 cells/mm^3^ with the exception of all HTLV-1/HIV-1-coinfected patients who were offered ART irrespective of CD4+ cell count [25,26] and this may have accounted for less deaths among these patients as compared to the HIV-1/Hepatitis B-coinfected patients in the study. Studies have suggested that combination ART therapy normalizes survival time in HIV-1/HTLV-1-coinfected patients [39]. There were no HTLV-1-related diseases seen in our HIV-1/HTLV-1-coinfected study patients.

As HTLV-1 is endemic in the Caribbean, a strength of the HIV programme in Trinidad and Tobago is the offering of HTLV-1 testing to all persons diagnosed with HIV so that HIV-1/HTLV-1-coinfected individuals may be more closely monitored, as persons living with HTLV-1 have an increased risk of premature death compared to those who are HTLV-1-uninfected [5]. The follow-up of the HIV-1/HTLV-1-coinfected patients is integrated into the HIV programme and they are followed up at their regular visits for antiretroviral therapy (ART) collection and for blood investigations. As there are no antivirals available to treat HTLV-1 and there is no available vaccine, the patients are offered counselling, which is similar for patients who are HIV mono-infected to reduce transmission of both HIV and HTV-1 via breastfeeding [10,11], via sexual intercourse [12] and parenterally via sharing of needles/syringes and transfusion of infected blood and blood products [2]. Patients are advised that they should consider the use of condoms to prevent sexual transmission of HTLV-1 [40], they should not donate blood, body organs, semen or other tissue [40], they should not share needles/syringes with anyone and pregnant women should try to refrain from breastfeeding infants [40,41] and their infants should be given formula milk, which is offered free of charge as part of the prevention of mother-to-child transmission (PMTCT) of the HIV programme in Trinidad and Tobago. St Lucia has integrated universal HTLV-1 screening of all pregnant women and replacement infant formula as part of the elimination of mother-to-child transmission of HIV and syphilis in the Caribbean [41] and it is expected that the rest of the Caribbean will follow this lead. T&T already has a policy in place for universal screening of blood donors for HTLV-1 [42]. It is recommended that persons coinfected with HIV-1/HTLV-1 in the study should be counselled to avoid condomless sex [40], should undergo regular screening and treatment of STIs (which may be asymptomatic) and their sexual partners and family members should be screened for HTLV-1 [43]. In addition, targeted HTLV-1 screening should be offered to persons who attend STI clinics, high-risk individuals such as sex workers, MSM and those with multiple sex partners and a once in a lifetime HTLV-1 screening should be offered for persons infected with hepatitis B and C [43].

Studies have shown that HIV-1/Hepatitis B-coinfected patients have a more rapid progression of liver disease and an increased risk of liver-associated mortality compared to hepatitis B mono-infected individuals [44,45]. There were 10 deaths in the MSM patients coinfected with HIV-1/Hepatitis B and 15 deaths in the heterosexual patients coinfected with HIV-1/Hepatitis B (*p* = 0.34). The causes of death among the MSM patients included six patients with AIDS-related complications, one patient with chronic renal failure, one patient due to homicide and two patients with unknown causes. The causes of death in the 15 HIV-1/Hepatitis B-coinfected heterosexual males included 10 patients with AIDS-related complications, one patient with chronic renal failure, one patient due to neurosyphilis, one patient due to myocardial infarction and two patients with unknown causes. In the study, no patients died due to complications of chronic hepatitis. Of the 1822 heterosexual males, there were 496 (27.2%) deaths as compared to the 1255 MSM study patients with 89 (7.1%) deaths (*p* = 0.02). Further analysis of the causes of death among the heterosexual males to determine possible contributory factors is proposed.

In addition, the first line ART regimen for the treatment of HIV in T&T is a combination of efavirenz/tenofovir/emtricitabine and the combination of tenofovir/emtricitabine is an effective treatment for persons coinfected with hepatitis B, this may have reduced the number of deaths related to the complications of chronic hepatitis B in the study. A hepatitis B vaccine is available free of charge as part of the expanded programme on immunization to all children living in T&T and the MRFTT offers the hepatitis B vaccine at no cost to all PLHIV attending the clinic and treatment is offered to all HIV-1/HBsAg coinfected patients (as part of ‘treat all’) with the aim of hepatitis B elimination by 2030.

The study has some limitations, there is a stigma associated with being a member of the MSM community in T&T so some study patients may not have disclosed their true sexual orientation resulting in an underestimate of the number of MSM attending the HIV Clinic. Another limitation of the study is the uncertainty of the duration of HTLV-1 infection and hepatitis B infection in the HIV-1/HTLV-1 and HIV-1/HBsAg-coinfected patients and it is unclear if HTLV-1 or hepatitis B preceded HIV-1 infection or vice versa as studies have shown that patients living with HIV-1 who are coinfected with either of these viruses may have a worse prognosis [14,44,45]. The study determined the prevalence of HTLV-1 and HBsAg positivity among MSM and heterosexual males attending one HIV Clinic in Trinidad and this may not be generalizable to the community of MSM and heterosexual males living with HIV in T&T.

This is the first study from the Caribbean to document the prevalence of HTLV-1 and HBsAg positivity among MSM attending the HIV Clinic in Trinidad. Despite the availability of an efficacious vaccine, there is a prevalence of hepatitis B of 5.1% among MSM living with HIV in the study; therefore, programmes to increase health literacy, screening and immunization among this key population group are urgently needed.

## Figures and Tables

**Table 1 viruses-16-01169-t001:** Baseline characteristics of the study population at initial visit (April 2002–October 2023).

Descriptive Statistic	Homosexual/Bisexual Males (*n* = 1255)	Heterosexual Males (*n* = 1822)	*p*-Value
Median Age	40 years	52 years	1.05
Age Range	19–85 years	18–94 years	
Ethnicity			0.14
African	681 (54.3%)	1198 (65.8%)	
East Indian	144 (11.5%)	139 (7.6%)	
Mixed	283 (22.5%)	282 (15.5%)	
Other/Caucasian/Chinese//Lebanese	31 (2.5%)	10 (0.5%)	
Not stated	116 (9.2%)	193 (10.6%)	
AIDS-defining illnesses	144 (11.8%)	441 (23.4%)	0.12
Medan CD4+ count (cells/mm^3^)	340	226	0.4
Median HIV-1 viral load (copies/mL)	25,968	45,427	<0.01

**Table 2 viruses-16-01169-t002:** Prevalence of HTLV-1 and HBsAg coinfection by ethnicity among MSM and heterosexual males attending a large HIV Treatment Centre in Trinidad.

Descriptive Statistic	Homosexual/Bisexual Males (*n* = 1255)	Heterosexual Males (*n* = 1822)	*p*-Value
HTLV-1 Seroprevalence			
Entire study population	1.67% (21/1255)	2.58% (47/1822)	0.12
African	2.05% (14/681)	3.01% (36/1198)	0.28
East Indian	2.08% (3/144)	1.44% (2/139)	1
Mixed	1.41% (4/283)	3.19% (9/282)	0.26
HBsAg positivity			
Entire study population	5.10% (64/1255)	3.79% (69/1822)	0.1
African	6.61% (45/681)	4.34% (52/1198)	0.04
East Indian	3.47% (5/144)	3.60% (5/139)	1
Mixed	4.59% (13/283)	4.24% (12/283)	1

**Table 3 viruses-16-01169-t003:** Comparison of HIV-1/HTLV-1 coinfection among heterosexual males vs. MSM patients at initial visit (April 2002–October 2023).

Descriptive Statistic	Homosexual/Bisexual (*n* = 21)	Heterosexual (*n* = 47)	*p*-Value
Median Age	34 years	60 years	0.13
Age Range	21–68 years	35–93 years	
Ethnicity			
African	14 (66.7%)	36 (76.6%)	0.25
East Indian	3 (14.3%)	2 (4.3%)	
Mixed	4 (19.0%)	9 (19.1%)	
AIDS-defining illnesses	4 (19.0%)	17 (36.2%)	0.06
Medan CD4+ count (cells/mm^3^)	210	257	0.34
Median HIV-1 viral load (copies/mL)	46,238	51,250	0.25

**Table 4 viruses-16-01169-t004:** Comparison of HIV-1/HBsAg coinfection among heterosexual males vs. MSM patients at initial visit (April 2002–October 2023).

Descriptive Statistic	Homosexual/Bisexual (*n* = 64)	Heterosexual (*n* = 69)	*p*-Value
Median Age	36 years	49 years	0.67
Age Range	22–73 years	22–77 years	
Ethnicity			
African	45 (70.3%)	52 (75.4%)	0.32
East Indian	5 (7.8%)	5 (7.2%)	
Mixed	13 (20.3%)	12 (17.4%)	
Other/Chinese	1 (1.6%)		
AIDS-defining illnesses	14 (21.9%)	24 (34.8%)	0.08
Medan CD4+ count (cells/mm^3^)	353	208	<0.01
Median HIV-1 viral load (copies/mL)	34,376	76,198	<0.01

## Data Availability

All data analyzed or generated during this study are included in the manuscript.

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
