# Peer review of "Prevalence of HTLV-1 and Hepatitis B Surface Antigen (HBsAg) Positivity among MSM Attending a Large HIV Treatment Centre in Trinidad"

_viruses, 2024, doi:10.3390/v16071169_

Round 1
Reviewer 1 Report
Comments and Suggestions for Authors
Edwards and colleagues conducted a very interesting study to assess the prevalence of Hepatitis B and HTLV-1 in a cohort of MSM living with HIV in Trinidad and Tobago. The findings of this study may inform policies targeting these infections.
My main comment is about the comparison between HIV/HBV to HIV/HTLV. It would be more informative if authors could compare these groups to MSM living with HIV single infection or heterosexual individuals (stratified by gender) living with HIV (monoinfection).
A strength of the HIV programme in Trinidad and Tobago is offering test for HTLV. This is not common in other countries, so it would be interesting if the authors could highlight and expand on this in the discussion session (lines 210-224). It would be interesting to know what is done with those who are diagnosed with HTLV-1. Are they offered counselling? Are there awareness campaigns /informative material? Are they offered follow-up? If so, how is the follow-up? Is it integrated into HIV programme? Are healthcare workers trained for HTLV? Are there guidelines for HIV/HTLV and HIV/HBV co-infection?
Minor details:
Line 24 HTLV-1 was the first human retrovirus to be identified.
Lines 28-32. It is important to highlight the association between HTLV-1 and increased risk of all-cause mortality (Schierhout et al). It is also questioned if 95% will indeed be asymptomatic, as, in addition to increased mortality there are many diseases associated to HTLV-1, impact on co-infections and socio-economic impact.
Lines63-64 It would be interesting to know what 12,000 PLHIV means in terms of prevalence.
Methods.
2.3 Data Analysis.
Instead of only reporting the statistical tests used I would recommend adding an explanation of the analysis that were performed. For example, used chi-square to assess… Mann Whitney was used to compare…
Results.
As HTLV testing comprises a diagnostic algorithm based on screening and confirmation it would be important to report if all samples that were reactive in ELISA were tested by WB. Was there any loss of follow-up (missing WB results)? It would be interesting to know the frequency of false positive results at screening test for HTLV and if there were Western Blot indeterminate/untyped samples (if so, how many).
Discussion
219-220 St Lucia and French West Indies have a programme to prevent vertical transmission. See ref below for ST Lucia
https://www.paho.org/en/documents/diagnosis-human-t-lymphotropic-virus-htlv-and-strategies-expand-htlv-screening-context
Author Response
Reviewer #1
Edwards and colleagues conducted a very interesting study to assess the prevalence of Hepatitis B and HTLV-1 in a cohort of MSM living with HIV in Trinidad and Tobago. The findings of this study may inform policies targeting these infections.
My main comment is about the comparison between HIV/HBV to HIV/HTLV. It would be more informative if authors could compare these groups to MSM living with HIV single infection or heterosexual individuals (stratified by gender) living with HIV (monoinfection).
Response:
The aim of the study was to determine the prevalence of HTLV-1 and HBsAg positivity at initial enrolment among MSM and heterosexual males attending a large HIV Clinic in Trinidad. Chart reviews were conducted between January 02-15, 2024, among self-identified MSM and a comparative group of randomly selected self-identified heterosexual males where sociodemographic, clinical and laboratory data were collected and analysed using SPSS Version 25. To determine if there are differences in the prevalences of HTLV1 and Hepatitis B between self-identified heterosexual males and MSM, a power of 90%, a significance level of 0.05 and an estimated prevalence of 3% for Hepatitis B for heterosexual males (Diez-Padrisa et al 2013) was used. It was calculated that 1822 Heterosexual Males would be required for the comparator group to ensure that the study was adequately powered to detect a true difference if it exists.
A strength of the HIV programme in Trinidad and Tobago is offering test for HTLV. This is not common in other countries, so it would be interesting if the authors could highlight and expand on this in the discussion session (lines 210-224).
Response
As HTLV-1 is endemic in the Caribbean, a strength of the HIV programme in Trinidad and Tobago is the offering of HTLV-1 testing to all persons diagnosed with HIV so that HIV-1/HTLV-1 coinfected individuals may be more closely monitored as persons living with HTLV-1 have an increased risk of premature death compared those who are HTLV-1 uninfected (Scherhout et al). The follow-up of the HIV-1/HTLV-1 coinfected patients is integrated into the HIV programme and they are followed up at their regular visits for antiretroviral therapy (ART) pick up and for blood investigations. The patients are offered counselling which is similar for patients who are HIV mono-infected to reduced transmission of both HIV and HTV-1 via breastfeeding [9, 10], via sexual intercourse [11] and parenterally via sharing of needles/syringes and transfusion of infected blood and blood products [2]. Patients are advised that they should consider the use of condoms to prevent sexual transmission of HTLV-1, they should not donate blood, body organs, semen or other tissue, they should not share needles/syringes with anyone and pregnant women should try to refrain from breastfeeding infants and the infants should be given formula milk which is offered free of charge as part of the prevention of mother to child transmission (PMTCT) of HIV programme in Trinidad and Tobago. St Lucia has integrated universal HTLV-1 screening of all pregnant women as part of the elimination of mother to child transmission of HIV and syphilis in the Caribbean [PAHO] and it is expected that the rest of the Caribbean will follow this lead.
It would be interesting to know what is done with those who are diagnosed with HTLV-1. Are they offered counselling? Are there awareness campaigns /informative material? Are they offered follow-up? If so, how is the follow-up? Is it integrated into HIV programme?
Response
The follow-up of the HIV-1/HTLV-1 coinfected patients is integrated into the HIV programme and they are followed up at their regular visits for antiretroviral therapy (ART) pick up and for blood investigations.. The patients are offered counselling which is similar for patients who are HIV mono-infected. and advised that they should share the information with their physicians, try to refrain from breastfeeding infants and they should consider the use of condoms to prevent sexual transmission of HTLV-1 especially if the person is in a mutually monogamous relationship, they should not donate blood, body organs, semen or other tissue and they should not share needles/syringes with anyone.
Are healthcare workers trained for HTLV? Are there guidelines for HIV/HTLV and HIV/HBV co-infection?
Response
A few healthcare workers at the Medical Research Foundation of Trinidad and Tobago (MRFTT) have over 25 years experience working/conducting research on patients with HTLV-1. The MRFTT uses the US HIV Clinical Guidelines for the management of HIV/HBC coinfection but unfortunately no treatment is recommended for persons with asymptomatic HTLV-1 infection. HIV-1/HTLV-1 coinfected patients are screened for comorbidities and coinfections (STIs, HBsAg, parasite infections ec). Patients with symptoms and signs suggestive of ATLL are referred to the haematologist and those with symptoms/signs suggestive of HAM/TSP are referred to the neurologist/physical therapist,
Minor details:
Line 24 HTLV-1 was the first human retrovirus to be identified.
Response:
This was corrected
Lines 28-32. It is important to highlight the association between HTLV-1 and increased risk of all-cause mortality (Schierhout et al). It is also questioned if 95% will indeed be asymptomatic, as, in addition to increased mortality there are many diseases associated to HTLV-1, impact on co-infections and socio-economic impact.
Response
Globally, it is estimated that approximately 5- 10 million people may be infected with HTLV-1 [3] and a systematic review and metaanalysis conducted by Schierhout et al [ ] showed that compared with HTLV-1 negative persons, the adjusted risk of death from any cause was higher in persons infected with HTLV-1 (RR 1·57, 95% CI 1·37–1·80). Approximately 5%-10% of infected patients develop a number of HTLV-1 associated diseases including adult T-cell leukemia/lymphoma (ATLL) [1,5], tropical spastic paraparesis/HTLV-1 associated myelopathy (TSP/HAM) [3, 6], infective dermatitis [7] and uveitis [8].
Lines63-64 It would be interesting to know what 12,000 PLHIV means in terms of prevalence.
Response
UNAIDS estimates that Trinidad and Tobago has an adult (aged 15-45 years ) with a HIV prevalence rate of 1%
Methods.
2.3 Data Analysis.
Instead of only reporting the statistical tests used I would recommend adding an explanation of the analysis that were performed. For example, used chi-square to assess… Mann Whitney was used to compare…
Response
Data were analyzed using various statistical methods to explore relationships and differences within the dataset. Measures of central tendency (mean and median) summarized numerical variables like age, CD4+ count, and HIV-1 viral load.
Chi-square tests assessed associations between categorical variables such as infection type (HIV-1/HBsAg vs. HIV-1/HTLV-1), ethnicity, sex, sexual orientation (heterosexual vs. homosexual/bisexual), and AIDS-defining infections. Fisher’s Exact Test was used when expected frequencies were low.
The Mann-Whitney U Test (Wilcoxon Rank Sum Test) compared numerical variables between two groups, providing a non-parametric alternative for data that did not follow a normal distribution. This test was used to compare variables such as age, CD4+ count, and HIV-1 viral load across different groups based on sexual orientation and infection type.
Results.
As HTLV testing comprises a diagnostic algorithm based on screening and confirmation it would be important to report if all samples that were reactive in ELISA were tested by WB. Was there any loss of follow-up (missing WB results)? It would be interesting to know the frequency of false positive results at screening test for HTLV and if there were Western Blot indeterminate/untyped samples (if so, how many).
Response:
All patients entering care at the HIV Clinic MRFTT undergo screening tests for HTLV-1 antibodies by ELISA and all ELISA positive tests are confirmed by western blot. There were three samples that were positive by ELISA and indeterminate by western blot (WB). These patients were re-screened by ELISA and WB approximately 6-12 months after initial sample collection and still remained WB indeterminate, so these patients were excluded from the study. As patients visit the clinic at 3-6 month intervals for ART pick up and blood investigations, there were no missing WB results.
Discussion
219-220 St Lucia and French West Indies have a programme to prevent vertical transmission. See ref below for ST Lucia
https://www.paho.org/en/documents/diagnosis-human-t-lymphotropic-virus-htlv-and-strategies-expand-htlv-screening-context
Response:
Mother to child transmission of HTLV-1 is a major mechanism of transmission [9, 10], St Lucia has integrated universal HTLV-1 screening of all pregnant women as part of the elimination of mother to child transmission of HIV and syphilis in the Caribbean. In T&T, all pregnant women are tested for HIV, those testing positive are offered ART (if not already on ART) and after delivery, breastfeeding is not recommended and the government provides all women living with HIV who deliver a baby, free formula milk for one year.
Reviewer 2 Report
Comments and Suggestions for Authors
This manuscript submitted by Edwards et al. describes the prevalence of HTLV-1 and HBV among HIV carriers in Trinidad. Unfortunately, there are so many serious problems and inconsistencies. The quality of this paper is quite low.
1) There is no list for abbreviations for example MSM (which is a keyword) PLHIV, MRFTT.... These abbreviations are not universally used and this might make it difficult to understand
2) In lines 68&69 the investigator mentioned that MSM is disproportionately affected by HIV but there is, no study to back it up the writeup apart from, in discussion line 186 where, the word ''thought'' was used.
3) Based on lines 72&73 his study aims to understand the burden among this population group (MSM) and implement prevention and control strategies. Why then, were measures that are not specific to this population (discouraging breastfeeding in seropositive women and free milk supply) added to the recommendation? I believe other recommendations specific to this population would have been added if, the study included the risk factors specific to them.
3b) lines 218-224. Since breastfeeding was not/cannot be identified, as a risk factor/the main risk factor among this population, it should not have been added in the discussion/conclusion as a recommendation based on the study but rather, in the introduction.
4) In the discussion lines 196-199 the investigator, mentioned prevalence among MSM and other homosexuals in Abuja and India but, did not mention these studies at all in the introduction/literature review. The literature review regarding this specific population was poor.
5) Based on lines 83-89 all clients, who positive for HTLV-1 between 2002-2012 were re-screened with HTLV-1/2 Ab Diapro ELISA, this is commendable because none of the clients was lost to follow-up. However;
- when was the re-screening done? 2013-2023?
- why was the re-screening done?
- Is HTLV-1/2 Ab Diapro ELISA better than the Vironostika HTLV-1 ELISA? if it is, was there a reason only previously HTLV-1/HIV positive patients were retested and not all HIV patients? OR is it to differentiate HTLV-1 and HTLV-2 antibodies? if yes based, on this article western blot was used to do that already.
- If the investigator does not know the reason, he/she should have included that in the writeup.
6) There is no need to include Females in your exclusion criteria since they are not part of the study population
7) lines 134-137 (the sentences after ''males and 4757 (45.6%) females'') can be excluded or included in the supplementary results or appendix since there is no real bearing on the main results (females were included in this grouping)
8) I don't think the type of study and study design was clearly stated in the methodology.
Minor comments
1) Line 25: Southeast is wrong. Southwest is correct.
2) Line 83: Here is the first appearance of ELISA
Comments on the Quality of English Languagenone
Author Response
Reviewer #2
This manuscript submitted by Edwards et al. describes the prevalence of HTLV-1 and HBV among HIV carriers in Trinidad. Unfortunately, there are so many serious problems and inconsistencies. The quality of this paper is quite low.
- There is no list for abbreviations for example MSM (which is a keyword) PLHIV, MRFTT.... These abbreviations are not universally used and this might make it difficult to understand
Response
A list of abbreviations was put in the manuscript
ART – antiretroviral therapy
ELISA - enzyme-linked immuno-assay
HBsAg- hepatitis B surface antigen
HTLV-1- human T-lymphotropic virus type 1
MSM – Men who have sex with men
PCP – pneumocystis pneumonia
PLHIV – persons living with HIV
PMTCT – prevention of mother to child transmission
MRFTT- Medical Research Foundation of Trinidad and Tobago
STI – sexually transmitted infection
TB - tuberculosis
T&T – Trinidad and Tobago
WB – western blot
2) In lines 68&69 the investigator mentioned that MSM is disproportionately affected by HIV but there is, no study to back it up the writeup apart from, in discussion line 186 where, the word ''thought'' was used.
Response:
HIV-1, Hepatitis B and HTLV-1 have similar risk factors and shared routes of transmission and MSM are disproportionately affected by HIV due to increased susceptibility of the rectal mucosa to HIV acquisition and transmission [28, 29], increased number of lifetime partners [28.29], role versatility [29], high risk sexual networks [29], disinhibited substance use [30], condomless anal sex [28,29], frequent asymptomatic sexually transmitted infections (STIs) with resulting inflammation and ulceration [31,32], internalized stigma that may result in depression and other mental health disorders [29, 30] and perceived and experienced discrimination by healthcare providers [33]
3) Based on lines 72&73 his study aims to understand the burden among this population group (MSM) and implement prevention and control strategies. Why then, were measures that are not specific to this population (discouraging breastfeeding in seropositive women and free milk supply) added to the recommendation? I believe other recommendations specific to this population would have been added if, the study included the risk factors specific to them.
Response
The study was retrospective and compared the HTLV-1 and HBsAg positivity among MSM and heterosexual males, data on risk factors was not collected. It is recommended that persons coinfected with HIV-1/HTLV-1 in the study should be counselled to avoid condomless sex [40], should undergo regular screening and treatment of STIs (which may be asymptomatic) and their sexual partners and family members should be screened for HTLV-1 [43]. In addition, targeted HTLV-1 screening should be offered to persons who attend STI clinics, high risk individuals such as sex workers, MSM and those with multiple sex partners and once in a life-time screening HTLV-1 screening should be offered for persons infected with hepatitis B and C [43]
3b) lines 218-224. Since breastfeeding was not/cannot be identified, as a risk factor/the main risk factor among this population, it should not have been added in the discussion/conclusion as a recommendation based on the study but rather, in the introduction.
Response
Refraining from breastfeeding was removed as one of the recommendations in the discussion/conclusion but was mentioned as one of the factors used in the counselling of persons infected with HTLV-1.
Patients are advised that they should consider the use of condoms to prevent sexual transmission of HTLV-1 [40], they should not donate blood, body organs, semen or other tissue [40], they should not share needles/syringes with anyone and pregnant women should try to refrain from breastfeeding infants [40, 41]and their infants should be given formula milk which is offered free of charge as part of the prevention of mother to child transmission (PMTCT) of HIV programme in Trinidad and Tobago. St Lucia has integrated universal HTLV-1 screening of all pregnant women and replacement infant formula as part of the elimination of mother to child transmission of HIV and syphilis in the Caribbean [41] and it is expected that the rest of the Caribbean will follow this lead.
4) In the discussion lines 196-199 the investigator, mentioned prevalence among MSM and other homosexuals in Abuja and India but, did not mention these studies at all in the introduction/literature review. The literature review regarding this specific population was poor.
Response
There are few reported studies of HIV-1/HBsAg coinfection in MSM living with HIV, a study conducted in 12 cities in India over the period 2012-2013 demonstrated a HBsAg prevalence of 8% [26] and a study conducted in Abuja and Lagos Nigeria among MSM and transgender women living with or at high risk for HIV showed the prevalence of HIV-1/HBV coinfection was 6%.
5) Based on lines 83-89 all clients, who positive for HTLV-1 between 2002-2012 were re-screened with HTLV-1/2 Ab Diapro ELISA, this is commendable because none of the clients was lost to follow-up. However;
- when was the re-screening done? 2013-2023?
- why was the re-screening done?
- Is HTLV-1/2 Ab Diapro ELISA better than the Vironostika HTLV-1 ELISA? if it is, was there a reason only previously HTLV-1/HIV positive patients were retested and not all HIV patients? OR is it to differentiate HTLV-1 and HTLV-2 antibodies? if yes based, on this article western blot was used to do that already.
- If the investigator does not know the reason, he/she should have included that in the writeup.
Response
From 2002 to 2012, the Vironostika HTLV-1 enzyme-linked immuno-assay (ELISA) (Vironostika, Organon Teknika, Durham DC) was used as the screening ELISA test, which was then discontinued and followed by the use of the HTLV-1/2 Ab Diapro ELISA (Dia.Pro, Italy). No western blot confirmation was done on the HTLV-1 positive samples over the period 2002-2018 and sera were frozen and stored, awaiting western blot confirmation. All clients with ELISA results seropositive for HTLV-1 over the period 2002-2012 were re-screened with the HTLV-1/2 Ab Diapro ELISA (Dia.Pro, Italy) so that there was consistency of the ELISA screening. All positive ELISA samples over the period April 2002-October 2023 were confirmed in batches from 2018 onwards using the western blot (MP Diagnostic TM HTLV BLOT 2.4 Western Blot Assay, MP Biomedicals Asia Pacific Pte. Ltd) which is able to differentiate antibodies to HTLV-1 and HTLV-2.
Resources were not available to re-screen the entire clinic population with the with the HTLV-1/2 Ab Diapro ELISA (Dia.Pro, Italy) and a number of patients had already died and were not available for re-screening.
6) There is no need to include Females in your exclusion criteria since they are not part of the study population
Response
Females were removed from the exclusion criteria
7) lines 134-137 (the sentences after ''males and 4757 (45.6%) females'') can be excluded or included in the supplementary results or appendix since there is no real bearing on the main results (females were included in this grouping)
Response
lines 134-137 (the sentences after ''males and 4757 (45.6%) females'' were removed
8) I don't think the type of study and study design was clearly stated in the methodology.
Response
The study was a retrospective chart review study
Minor comments
- Line 25: Southeast is wrong. Southwest is correct.
Response : Corrected
2) Line 83: Here is the first appearance of ELISA
Response - Corrected : the enzyme-linked immuno-assay (ELISA)
Round 2
Reviewer 2 Report
Comments and Suggestions for Authors
none
Author Response
No comments/suggestions from Reviewers for improvement